# Multiple *pkd* and *piezo* gene family members are required for atrioventricular valve formation

Thomas Juan [1,2,3] ✉, Agatha Ribeiro da Silva[1,2,5], Bárbara Cardoso[1,2,5], SoEun Lim [1,2], Violette Charteau[1,2,4] & Didier Y. R. Stainier [1,2,3] ✉

Cardiac valves ensure unidirectional blood flow through the heart, and altering their function can result in heart failure. Flow sensing via wall shear stress and wall stretching through the action of mechanosensors can modulate cardiac valve formation. However, the identity and precise role of the key mechanosensors and their effectors remain mostly unknown. Here, we genetically dissect the role of Pkd1a and other mechanosensors in atrioventricular (AV) valve formation in zebrafish and identify a role for several *pkd* and *piezo* gene family members in this process. We show that Pkd1a, together with Pkd2, Pkd1l1, and Piezo2a, promotes AV valve elongation and cardiac morphogenesis. Mechanistically, Pkd1a, Pkd2, and Pkd1l1 all repress the expression of *klf2a* and *klf2b*, transcription factor genes implicated in AV valve development. Furthermore, we find that the calcium-dependent protein kinase Camk2g is required downstream of Pkd function to repress *klf2a* expression. Altogether, these data identify, and dissect the role of, several mechanosensors required for AV valve formation, thereby broadening our understanding of cardiac valvulogenesis.

Congenital heart disease is the most common congenital disorder in newborns, with an incidence of approximately 1% of live births[1]. The most frequent congenital heart defects are cardiac valve abnormalities, which often originate from early developmental defects and are caused by mutations in genes conserved across vertebrates[2]. Cardiac valves are flexible structures that sit at the boundaries between the chambers. Their malfunction, leading to phenotypes such as regurgitation and stenosis, is one of the primary causes of heart failure[3]. Thus, a better understanding of the causes of congenital valve disorders will help address a major public health issue.

The zebrafish is a model of choice to study valve development because of its fast organogenesis and ease of optical approach implementation[4]. The zebrafish heart is two-chambered and comprises three types of valves: the sinus venosus, atrioventricular (AV), and

outflow tract (OFT) valves[5]. Cardiac valves differentiate from endocardial cells, which line the lumen of the heart, shortly after the heart starts beating. Their formation depends upon the hemodynamic forces applied on the endocardial layer[6,7]. Wall shear stress (WSS) and radial wall pressure, which induces endocardial cell stretching, are the main driving force for valve differentiation and elongation[8–12]. Although the genetics of flow-inducible genes have been extensively studied[13–17], relatively little is known about the upstream blood flow-sensing mechanisms.

Endocardial cells are sensitive to the subtle differences in force that accompany heart morphogenesis. Several families of mechanosensitive molecules are involved in vascular development, including ion channels, integrins, junctional complexes, G protein-coupled receptors, and the glycocalyx[14]. During cardiac valve formation

[1]Max Planck Institute for Heart and Lung Research, Department of Developmental Genetics, Bad Nauheim, Germany. [2]German Centre for Cardiovascular Research (DZHK), Partner Site Rhine-Main, Bad Nauheim, Germany. [3]Cardio-Pulmonary Institute (CPI), Bad Nauheim, Germany. [4]Present address: Institute for Molecules and Materials (IMM), Department of Biomolecular Chemistry, Radboud University, Nijmegen, The Netherlands. [5]These authors contributed equally: Agatha Ribeiro da Silva, Bárbara Cardoso. ✉e-mail: thomas.juan@mpi-bn.mpg.de; didier.stainier@mpi-bn.mpg.de

specifically, membrane-localized mechanosensitive channels, which are in direct contact with the blood flow, control valve elongation. Studies in zebrafish have shown that these mechanosensors, Pkd2 and Trpv4 in all cardiac valves and Piezo1 and Piezo2a in the OFT valves, tightly regulate *klf2a* expression[10,11,18], and that they subsequently control the endocardial and myocardial expression of other genes important for valvulogenesis[7,19–22]. In addition, it was also recently reported that the adenosine triphosphate (ATP)–dependent purinergic receptor pathway works in parallel to these channels as a blood flow mechanosensor to mediate AV valve formation[12]. However, the loss of any of these mechanosensors induces only mild phenotypes[10–12] compared with abnormal fluid flow conditions in the heart[6,23], suggesting the existence of additional mechanosensitive molecules required for valve development.

To test this hypothesis, we provide in this study genetic evidence that *pkd1a*, an orthologue of human *PKD1*, which is one of the most commonly mutated genes in Autosomal Dominant Polycystic Kidney Disease[24], is essential for AV valve elongation in zebrafish. We identify a synergistic contribution of Pkd1a and other Pkd family members, namely Pkd2 and Pkd1l1, in blood flow-driven valve development, independent of their role in cardiac looping. We further show that the Piezo family members Piezo1 and Piezo2a are also important in AV valve formation in addition to their role in OFT valve development and that they cooperate with Pkd1a during this process. Using high-resolution imaging, we find that these *pkd* genes control calcium influx as well as *klf2a* and *klf2b* expression in the AV endocardial cells. Using a *pkd* phenotypic rescue assay, we identify the Ca²⁺/calmodulin-dependent protein kinases Camk2g1 and Camk2g2 as intermediates of the Pkd mechanosensory response in AV valve formation. Taken together, our findings identify major regulators of AV valve development, thereby providing new potential therapeutic targets for congenital valve disorders.

## Results

### Pkd1a is required for atrioventricular valve development

Zebrafish *pkd1a* mutants display pericardial edema and lymphatic defects at 4 days post-fertilization (dpf) while retaining blood circulation[25]. To investigate the role of Pkd1a in AV valve development, independently of its function in cardiac and lymphatic morphogenesis, we first characterized the onset of its cardiac phenotype. To do so, we used a previously described mutant allele (hu5855)[25] as well as a newly generated mutant allele (bns507) (Fig. S1a). Both mutations lead to a premature termination codon and the disruption of the conserved polycystic cation channel domain (PCCD)[26]. We found that *pkd1a* mutants, of both alleles, exhibit wild-type-like morphology (Fig. 1a–c) and cardiac chamber size (Fig. 1d–f) at 78 hours post-fertilization (hpf), but clear defects at 102 hpf, when the cardiac chambers start to collapse (Fig. 1a′–f′) as shown before[25]. To investigate this phenotype, we first measured the diameter of the cardiac chambers during atrial diastole and systole (Fig. S1b–f) at 78 hpf, and observed no significant differences between *pkd1a* mutants and wild types (Fig. S1g, h), in contrast to what is observed in *tnnt2a* mutants, where the atrium is distended[27]. We then analyzed the blood flow profile across the AV canal in single beating cycles between 54 and 108 hpf, starting with wild-type embryos and larvae. We found that retrograde blood flow movements are almost absent at 102 hpf (Supplementary Movie 3), while strong at 54 and 78 hpf (Supplementary Movies 1, 2), and that following AV valve formation and the reduction of retrograde movements, a no-flow (i.e., no anterograde or retrograde flow) period starts to appear at 78 hpf (Fig. 1g and Supplementary Movie 2). Therefore, we selected 78 hpf to study the *pkd1a* mutant phenotype, which is the latest stage when *pkd1a* mutants display a wild-type-like cardiac chamber size. At this stage and at the AV canal, *pkd1a* mutant larvae, similarly to *klf2a; klf2b* double mutant larvae, display an increased incidence in retrograde blood flow and a decreased incidence in no-

flow (Fig. 1h and Supplementary Movie 4), while maintaining a normal heartbeat (Fig. S1i). We also analyzed AV valve morphology in live *pkd1a* mutant larvae and found that only 66% exhibited an elongated superior valve leaflet compared with 87% in wild-types (Fig. 1i–k). We then quantified the number of cells present in the superior valve leaflet and found no significant reduction in *pkd1a* mutants compared with wild type (Fig. 1l–n). Altogether, these data indicate that the Pkd1a function is required for early AV valve development.

### Pkd1a, 2, and 1l1 cooperate to promote AV valvulogenesis

Compared with blood flow-defective conditions, such as those observed in *tnnt2a* mutants[6,10,23], *pkd1a* mutants display only a mild AV valve phenotype (Fig. 1i–n), suggesting that other mechanosensory genes also regulate AV valve development. Pkd1a has been described to bind Pkd2[28], another Pkd member involved in AV valve development[10]. Moreover, Pkd2 has been shown to be complex with Pkd1l1 to control flow mechanosensation in the vertebrate left/right organizer[29,30]. We thus investigated whether mutant combinations for these three genes displayed a stronger valve phenotype than observed in *pkd1a* single mutants. For this analysis, we used a previously described *pkd2* mutant allele (tc321)[31] (Fig. S2a) and generated a frameshift allele for *pkd1l1* (bns394) (Fig. S2a′). Both mutations lead to a premature termination codon and the disruption of the PCCD domain[26]. We found that *pkd1l1*, but not *pkd2*, mRNA levels are upregulated in *pkd1a* mutants (Fig. S2b), suggesting genetic compensation via transcriptional adaptation (TA)[32]. *pkd1l1* mutant animals display the left/right (L/R) asymmetry phenotype previously reported in medaka and mice, with randomized cardiac looping[29,30], as do *pkd2* mutants[31] (Fig. S2c–g). Therefore, every mutant combination that involves *pkd2* or *pkd1l1* leads to randomized cardiac looping (Fig. S2c–g). As it was previously shown that cardiac looping defects can correlate with early AV valve defects[8], we sorted and analyzed only the larvae exhibiting wild-type cardiac looping for all subsequent experiments that involve *pkd2* or *pkd1l1* mutants. We show that these sorted double and triple *pkd* mutants display a wild-type-like cardiac chamber size (Fig. 2a–d and Fig. S2h, i), and beating frequency (Fig. S2j) at 78 hpf. When analyzing the AV flow profile of the *pkd* mutant combinations, we found that all double mutant combinations cause the same retrograde blood flow defects as *pkd1a* mutations do (Fig. 2e). However, *pkd* triple mutants exhibit an increased incidence in retrograde blood flow compared with *pkd1a; pkd2* double mutants (Fig. 2e and Supplementary Movie 5). Live imaging of the AV valve reveals that superior valve leaflet elongation is almost completely abolished in *pkd* triple mutant larvae (Fig. 2f–l). We also quantified the number of cells present in the superior valve leaflet and found no significant reduction in *pkd* double or triple mutants (Fig. 2m–s); in addition, valve luminal and abluminal cells are present in *pkd* triple mutants (Fig. 2t, u). These data show that Pkd1a, Pkd2, and Pkd1l1 cooperate during AV valve development and are required to promote AV valve elongation.

### Piezo family members are also involved in AV valvulogenesis

Another family of membrane-bound mechanosensors, the Piezo family, has been shown to control OFT valve formation[11,18]. However, their function in AV valve formation and their interaction with Pkd family members remain unexplored. Here, we have generated mutants for *piezo1* (bns340) (Fig. S3a) and *piezo2a* (bns367) (Fig. S3a′) that carry a deletion of two amino acids in the conserved PFEW domain, which is hypothesized to be involved in channel conductance or gating[33]. We selected in-frame alleles to try and minimize genetic compensation via TA[34], as described for some *piezo1* mutants[35]. We found that *piezo1* and *piezo2a* mRNA are enriched in the AV canal at 78 hpf (Fig. S3k, l) and that *piezo1* and *piezo2a* mRNA levels are unchanged in *pkd1a* mutants (Fig. S2b). As with *pkd2* and *pkd1l1* mutants (Fig. S2g), we found that mutants for *piezo2a*, but not *piezo1*, exhibit cardiac looping defects (Fig. S3c–g). We thus used larvae exhibiting wild-type cardiac looping

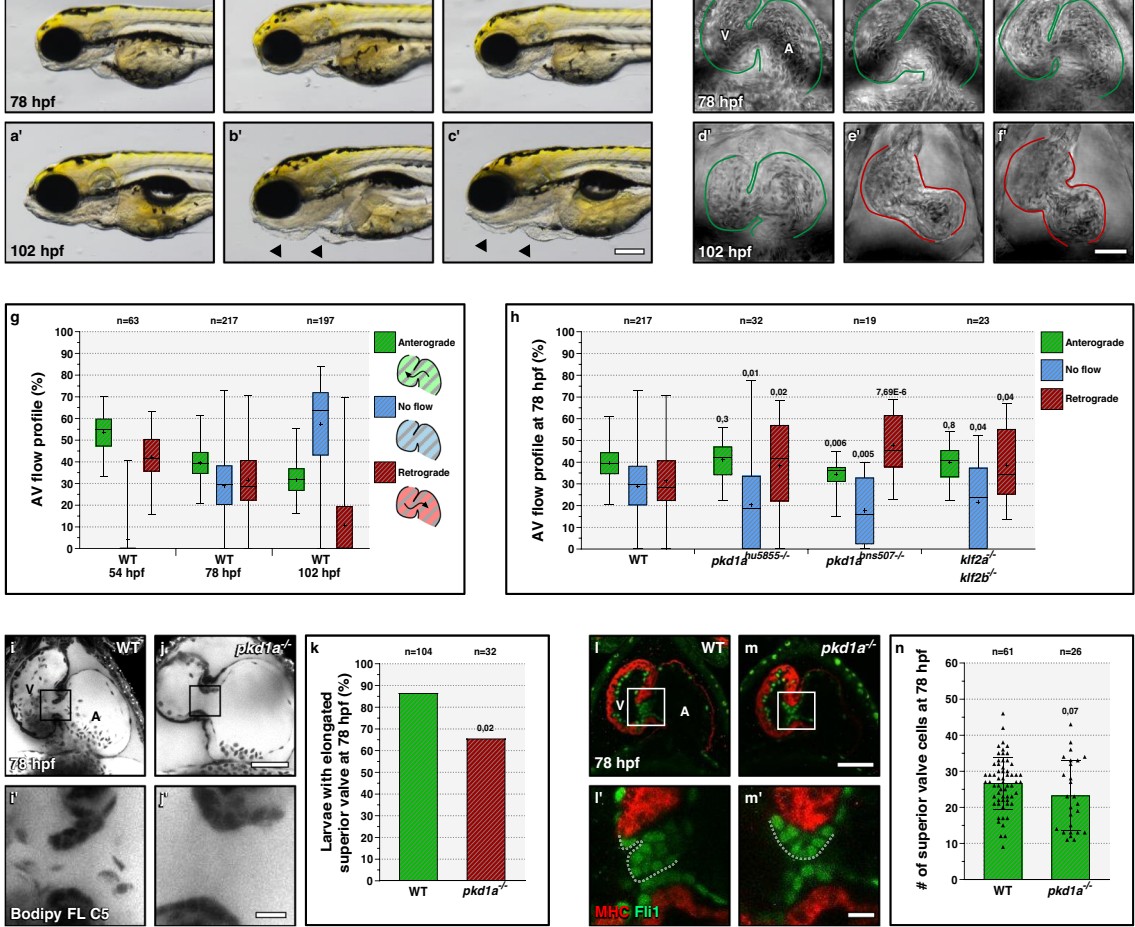

**Fig. 1 | *pkd1a* is required for atrioventricular valve development. a–c′** Wild-type (**a-a′**) like morphology of *pkd1a^hu5855^* (**b-b′**) and *pkd1a^bnsS07^* (**c-c′**) mutants at 78 but not 102 hpf; arrows point to craniofacial defects and pericardial edema. **d–f′** Brightfield images of 78 and 102 hpf wild-type (**d-d′**), *pkd1a^hu5855^* mutant (**e-e′**), and *pkd1a^bnsS07^* mutant (**f-f′**) hearts; one time point of a spinning disc movie at atrial diastole; green and red lines outline the heart. **g** AV flow profile of 54, 78, and 102 hpf wild-type zebrafish; average profile of three cardiac cycles per zebrafish. **h** AV flow profile of 78 hpf wild-type, *pkd1a^hu5855^* mutant, *pkd1a^bnsS07^* mutant, and *klf2a; klf2b* double mutant larvae; average profile of three cardiac cycles per zebrafish. **i–j′** Confocal imaging of 78 hpf wild-type (**i-i′**) and *pkd1a* mutant (**j-j′**) hearts stained with Bodipy FL C5-Ceramide; **i′, j′** are AV canal magnified images of **i, j. k** Superior

AV valve elongation of 78 hpf wild-type and *pkd1a* mutant larvae. **l–m′** Confocal imaging of 78 hpf wild-type (**l-l′**) and *pkd1a* mutant (**m-m′**) hearts immunostained for MHC and Fli1; **l′, m′** are AV canal magnified images of **l, l. n** Superior AV valve cell number in 78 hpf wild types and *pkd1a* mutants. Hearts imaged in ventral view, anterior to the top, ventricle (V) on the left, and atrium (A) on the right. The center of the box-and-whisker plots represents the median; each of the box-and-whiskers represents a quartile; "+" represents the mean in **g**, **h**. The center of the error bar represents the mean in **k**, **n**. Error bars indicate s.d in **n**. *P* values were calculated using a two-sided Student's *t*-test in **g**, **h**, **n**; a two-sided Fisher's exact test in **k**; and are relative to wild type. Scale bars: 200 μm in **a–c′**; 50 μm in **d–f′**, **i**, **j**, **l**, **m**; 10 μm in **i′**, **j′**, **l′**, **m′**. Source data are provided as a Source Data file.

for all subsequent experiments involving *piezo2a* mutants. Furthermore, we found that injecting *pkd1a* morpholinos (MOs) in *piezo2a* mutants enhances the cardiac looping defects, in comparison to injecting them into *piezo1* mutants (Fig. S3c–g), suggesting cooperation between *pkd1a* and *piezo2a* in regulating cardiac morphogenesis. We found that *piezo1* and *piezo2a* mutants exhibit a wild-type-like cardiac chamber size (Fig. 3a–d and Fig. S3h, i), and beating frequency (Fig. S3j) at 78 hpf. We analyzed the AV blood flow profile in *piezo1* and *piezo2a* mutants, and observed an increased incidence in retrograde movements, comparable with those observed in *pkd1a* mutants (Fig. 3e). Notably, *piezo2a; pkd1a* double mutants exhibit stronger blood flow defects than the single mutants, suggesting cooperation between *pkd* and *piezo* genes during AV valve development (Fig. 3e). Similarly, while *piezo1* mutants and *piezo2a* mutants display valve elongation defects, *piezo2a; pkd1a* double mutants exhibit stronger defects than the single mutants (Fig. 3f–j). We also quantified the number of cells present in the superior valve leaflet and found no significant reduction in *piezo1* mutants, *piezo2a* mutants, or *piezo2a; pkd1a* double mutants (Fig. 2k–o). Taken together, our data show that

Piezo1 and Piezo2a play an important role during AV valvulogenesis and that Piezo and Pkd family members cooperate in this process.

## *pkd* genes control calcium levels during valvulogenesis

Blood flow regulates intracellular calcium levels in the AV canal via the Trpv4 and Pkd2 mechanosensors to promote valvulogenesis[10]. We thus investigated whether Pkd1a, 2, and 1l1 also cooperate to control endocardial calcium levels, upstream of AV valve elongation. We first used the *fli1a:Gal4FF; UAS:GCamP6s* calcium sensor line[36] to measure endocardial calcium levels in 78 hpf wild-type larvae, during atrial diastole (Fig. S4a–c) and systole (Fig. S4a′–c′) and found that, as previously reported at 48 hpf[10], calcium levels are highest at the AV canal (Fig. S4d and Supplementary Movie 6). Moreover, we also observed higher calcium levels in the presumptive OFT valve area (Fig. S4c and Supplementary Movie 6). We found that both AV and OFT signals were enriched during diastole and systole, without significant differences between the cardiac states (Fig. S4c and Supplementary Movie 6). Strikingly, this calcium elevation was lost specifically in the AV canal in *tnnt2a* morphants, without a significant decrease in the other cardiac

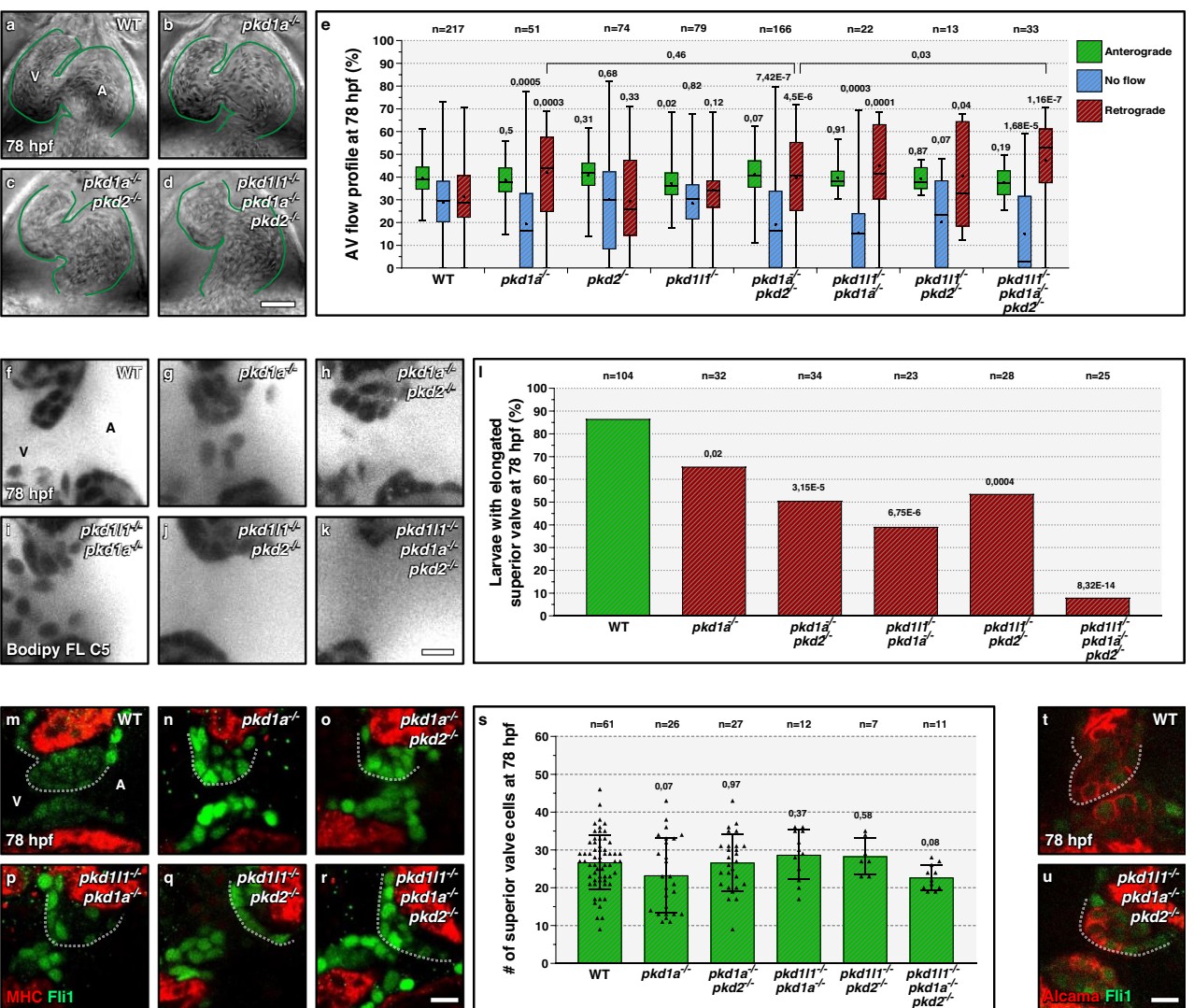

**Fig. 2 | *pkd1a*, *pkd2*, and *pkd1l1* cooperate during atrioventricular valve development. a–d** Brightfield images of 78 hpf wild-type (**a**), *pkd1a* mutant (**b**), *pkd1a; pkd2* double mutant (**c**), and *pkd* triple mutant (**d**) hearts; one time point of a spinning disc movie at atrial diastole; green lines outline the heart. **e** AV flow profile of 78 hpf wild-type, *pkd1a* mutant, *pkd2* mutant, *pkd1l1* mutant, *pkd* double mutant, and *pkd* triple mutant larvae; average profile of three cardiac cycles per zebrafish. **f–k** Confocal imaging of 78 hpf wild-type (**f**), *pkd1a* mutant (**g**), *pkd* double mutant (**h–j**), and *pkd* triple mutant (**k**) hearts stained with Bodipy FL C5-Ceramide; AV canal magnified images. **l** Superior AV valve elongation of 78 hpf wild-type, *pkd1a* mutant, *pkd* double mutant, and *pkd* triple mutant hearts. **m–r** Confocal imaging of 78 hpf wild-type (**m**), *pkd1a* mutant (**n**), *pkd* double mutant (**o–q**), and *pkd* triple mutant (**r**) hearts immunostained for MHC and Fli1; AV canal magnified images.

**s** Superior AV valve cell number in 78 hpf wild types, *pkd1a* mutants, *pkd* double mutants, and *pkd* triple mutants. **t, u** Confocal imaging of 78 hpf wild-type (**t**) and *pkd* triple mutant (**u**) hearts immunostained for Alcama and Fli1; 10/10 of the wild-type and 9/9 of the *pkd* triple mutant larvae display luminal and abluminal superior valve cells; AV canal magnified images. Hearts imaged in ventral view, anterior to the top, ventricle (V) on the left and atrium (A) on the right. The center of the box-and-whisker plot represents the median; each of the box-and-whiskers represents a quartile; "+" represents the mean in **e**. The center of the error bar represents the mean in **l**, **s**. Error bars indicate s.d in **s**. *P* values were calculated using a two-sided Student's *t*-test in **e**, **s**; a two-sided Fisher's exact test in **l**; and are relative to wild type except when a horizontal bar is used. Scale bars: 50 μm in **a–d**; 10 μm in **f–k**, **m–r**, **t**, **u**. Source data are provided as a Source Data file.

compartments (Fig. 4a–b', d, e–f', h and Supplementary Movie 8). *pkd* triple mutants mimic the *tnnt2a* morphant condition, with a loss of AV signal while maintaining an elevated calcium level in the ventricle compared with the atrium during atrial diastole (Fig. 4c, d, g, h and Supplementary Movie 7). We then analyzed the expression levels of valve-specific transcription factor genes in the heart. We selected *klf2a* and *klf2b*, whose loss is associated with AV valve defects, and which control a large set of endocardial and myocardial genes essential for valve formation[7,19,22]. We performed Fluorescence In Situ Hybridization (FISH) and observed apparent endocardial upregulation of *klf2a* (Fig. 5a–d and Fig. S5a–k) and *klf2b* (Fig. 5e–h and Fig. S5l–s) in *pkd* triple mutants (Fig. 5c, g), *pkd1a; pkd2* double mutants (Fig. S5f, n), and *piezo2a; pkd1a* double mutants (Fig. S5j, r) compared with *pkd1l1* mutants (Fig. 5b, f), *pkd2* mutants (Fig. Se, m), *pkd1a* mutants (Fig. 5i,

q), and wild types (Fig. 5a, e and Fig. S5a, d, h, l, p). Notably, the AV signal was stronger in these three conditions compared with the wild types and single mutants (Fig. 5d, h and Fig. S5g, k, o, s). We then performed RT-qPCR on dissected hearts to quantify this differential expression and observed an upregulation of *klf2a* (Fig. 5i) and *klf2b* (Fig. 5j) in *pkd* triple mutants, *pkd1a; pkd2* double mutants, and *piezo2a; pkd1a* double mutants compared with *pkd* single mutants and wild types. To investigate whether *klf2* upregulation could be responsible for the valve elongation phenotype we observed, we injected at the one-cell stage a plasmid encoding *UAS:klf2a-p2a-dTomato* in *nfatc1:Gal4FF* positive embryos (Fig. 5k, l). We then sorted the larvae negative for dTomato signal (Fig. 5k) from the positive ones (Fig. 5l) in the superior valve, and found that the positive ones displayed a higher frequency of valve elongation defects than the

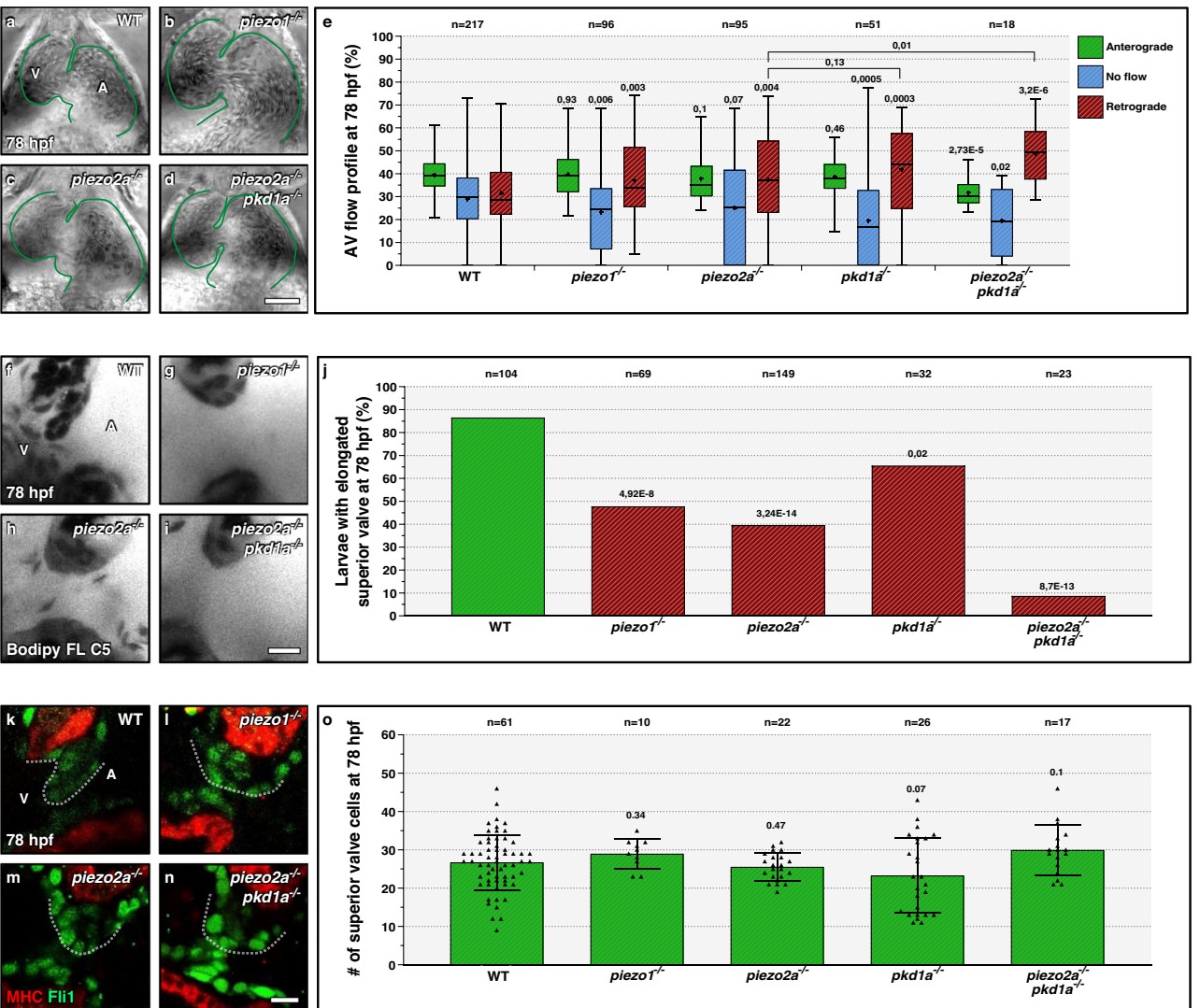

**Fig. 3 | *piezo1* and *piezo2a* promote atrioventricular valve development.**
**a–d** Brightfield images of 78 hpf wild-type (**a**), *piezo1* mutant (**b**), *piezo2a* mutant (**c**), and *piezo2a; pkd1a* double mutant (**d**) hearts; one time point of a spinning disc movie at atrial diastole; green lines outline the heart. **e** AV flow profile of 78 hpf wild-type, *piezo1* mutant, *pkd1a* mutant, *piezo2a* mutant, and *piezo2a; pkd1a* double mutant larvae; average profile of three cardiac cycles per zebrafish. **f–i** Confocal imaging of 78 hpf wild-type (**f**), *piezo1* mutant (**g**), *piezo2a* mutant (**h**), and *piezo2a; pkd1a* double mutant (**i**) hearts stained with Bodipy FL C5-Ceramide; AV canal magnified images. **j** Superior AV valve elongation of 78 hpf wild types, *piezo1* mutants, *pkd1a* mutants, *piezo2a* mutants, and *piezo2a; pkd1a* double mutants. **k–n** Confocal imaging of 78 hpf wild-type (**k**), *piezo1* mutant (**l**), *piezo2a* mutant (**m**), and *piezo2a; pkd1a* double mutant (**n**) hearts immunostained for MHC and Fli1;

AV canal magnified images; ventral views, anterior to the top, ventricle (V) on the left and atrium (A) on the right. **o** Superior AV valve cell number in 78 hpf wild types, *piezo1* mutants, *pkd1a* mutants, *piezo2a* mutants, and *piezo2a; pkd1a* double mutants. Hearts imaged in ventral views, anterior to the top, ventricle (V) on the left, and atrium (A) on the right. The center of the box-and-whisker plot represents the median; each of the box-and-whiskers represents a quartile; "+" represents the mean in **e**. The center of the error bar represents the mean in **j**, **o**. Error bars indicate s.d in **o**. *P* values were calculated using a two-sided Student's *t*-test in **e**, **o**; a two-sided Fisher's exact test in **j**; and are relative to wild type except when a horizontal bar is used. Scale bars: 50 μm in **a–d**; 10 μm in **f–i**, **k–n**. Source data are provided as a Source Data file.

negative ones. Together, these observations suggest that *pkd* and *piezo* genes cooperate to control endocardial calcium levels and negatively regulate the expression of essential AV transcription factor genes.

## Camk2g is an effector of the Pkd response upstream of *klf2a* expression

*pkd* genes control endocardial calcium levels and *klf2a* expression[10]. However, the intermediates of this signaling pathway in the AV canal remain unknown. In this context, the Ca2+/calmodulin-dependent (CaM) pathway has been reported to regulate *KLF2* mRNA levels in human umbilical vein endothelial cells (HUVECs)[37], and CaM protein Kinase II (CaMKII) phosphorylation is reduced in zebrafish *pkd2* morphants in two flow responsive tissues, the pronephros[38] and the left/

right organizer[39]. Therefore, we decided to test the role of CaMK proteins as potential intermediates of Pkd-driven control of valvulogenesis. In zebrafish, the most strongly expressed *camk* genes in embryonic endothelial cells are *camk2g1* and *camk2g2*[40], which we selected for our analysis. CaMKII kinases autophosphorylate at T287 upon calcium stimulation and Camk2g[T287D] variants are Constitutively Active (CA) even in the absence of calcium[38,41]. We found that retrograde blood flow movements (Fig. 6a–c) and AV valve elongation defects (Fig. 6d–f) of *pkd* triple mutants could be partially rescued by injecting mRNA encoding CA zebrafish (z) Camk2g1 or CA human (h) CAMK2G. CA zCamk2g1 overexpression also restored *klf2a* expression towards wild-type levels in *pkd* triple mutants (Fig. 6g–i), indicating that Camk2g1 is upstream of *klf2a* expression during valvulogenesis. To investigate the role of the Camk2g proteins in valve development,

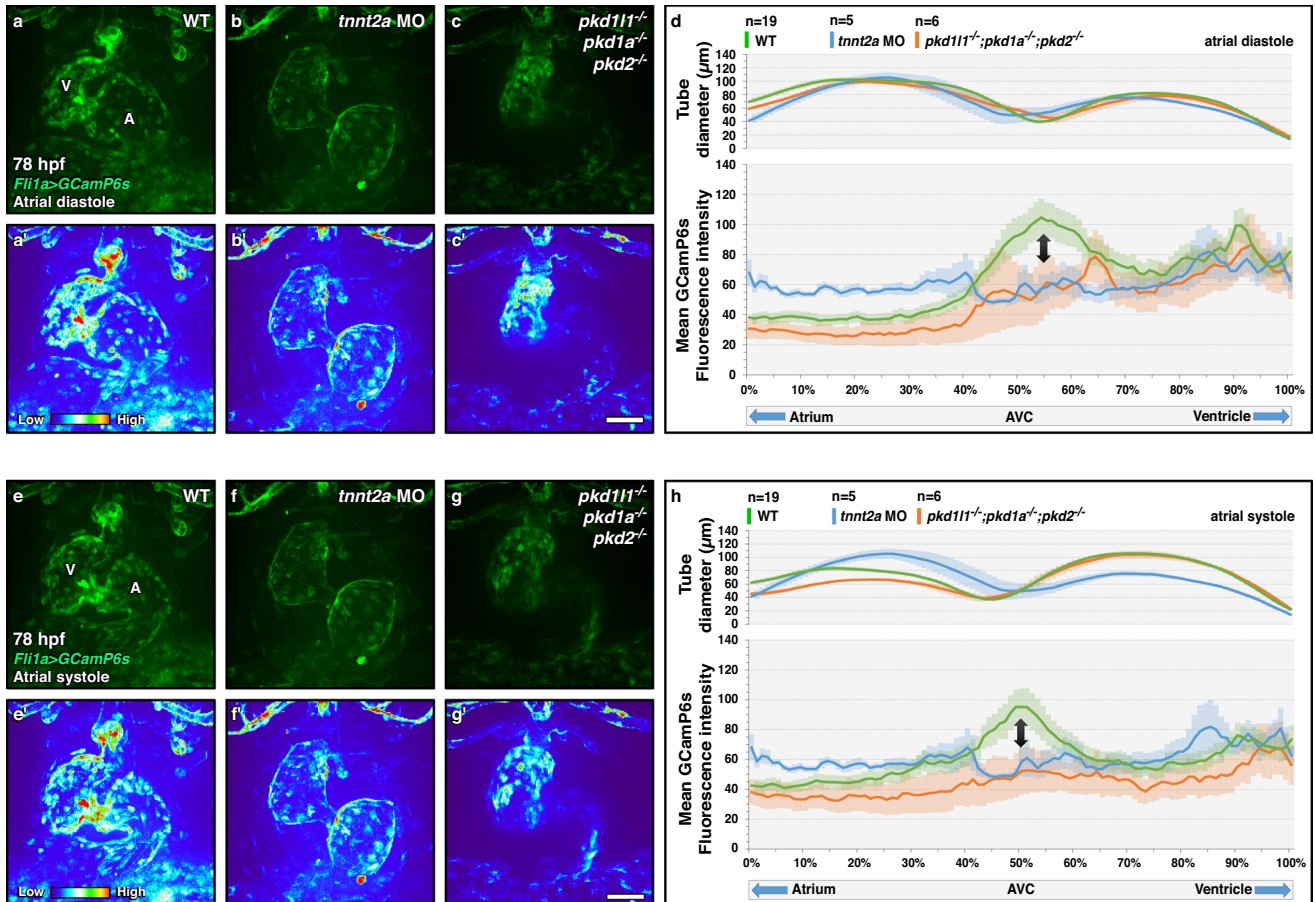

**Fig. 4 | Control of calcium levels in the atrioventricular canal by Pkd members.** **a–c**, **e–g** Confocal imaging of 78 hpf *fli1a:Gal4FF; UAS:GCaMP6s* wild-type (**a**, **a'**, **e**, **e'**), *tnnt2a* morphant (**b**, **b**, **f**, **f'**) and *pkd* triple mutant (**c**, **c'**, **g**, **g'**) hearts during atrial diastole (**a–c'**) and systole (**e–g'**); maximum projection of 4D-assembled spinning disc movies. **d**, **h** Diameter and endocardial calcium levels along the contracting

heart tube in 78 hpf wild-type, *tnnt2a* morphant, and *pkd* triple mutant larvae during atrial diastole (**d**) and systole (**h**); arrows point to the AV canal calcium peak. Hearts imaged in ventral view, anterior to the top, ventricle (V) on the left, and atrium (A) on the right. Error bars indicate SEM. Scale bars: 50 μm. Source data are provided as a Source Data file.

we generated frameshift alleles for *camk2g1* (*bns508*) (Fig. S6a) and *camk2g2* (*bns509*) (Fig. S6a'). Both mutations lead to a premature termination codon and the disruption of the conserved CaMKII Association Domain (CaMKII-AD). We analyzed valve-associated defects in *camk2g1; camk2g2* double mutants since *camk2g2* upregulation can compensate for the loss of *camk2g1*[42]. We found that *camk2g1; camk2g2* double mutants display a wild-type-like cardiac chamber size (Fig. S6b, c), and beating frequency (Fig. S6d), without cardiac looping defects (Fig. 6j, k) at 78 hpf. We then analyzed the AV flow profile as well as valve elongation and found that *camk2g1* mutants and *camk2g1; camk2g2* double mutants exhibit an increased incidence in retrograde blood flow (Fig. 6j–l) as well as valve elongation defects (Fig. 6m–o). *camk2g2* mutants also exhibit valve elongation defects (Fig. 6o) but no increased incidence in retrograde blood flow (Fig. 6i). Altogether, these findings suggest that Camk2g is an important modulator of valvulogenesis downstream of Pkd and upstream of *klf2a* expression (Fig. 6p).

## Discussion

Cardiac valve leaflet formation is sensitive to biomechanical forces generated by blood flow and cardiac muscle contraction[6,12,23]. Previous studies have shown that membrane-bound mechanosensitive ion channels are involved in valve formation[10,11,18]. However, the valve phenotypes associated with their loss are not as severe as conditions without blood flow[6,23]. Using live imaging of double and triple mutant combinations, we uncovered specific

mechanosensors with overlapping, or compensatory, functions in AV valve formation.

We first focused on members of the Pkd family described to interact with Pkd2, which is involved in cardiac valve development[10,11]. Pkd proteins form heteromeric complexes to mediate their membrane localization and function in multiple organs[43]. Consequently, mutations in partner pairs typically give rise to the same phenotypes[42], such as *pkd1l1* and *pkd2* in the left/right organizer[29,30,44]. In zebrafish, *pkd2* knockdown also shows the same morphological defects as *pkd1a* and *pkd1b* double knockdown[45], indicating that different heteromeric complexes can be formed between Pkd2 and several Pkd1-like proteins. We, therefore, decided to mutate the known heteromeric partners of Pkd2, namely Pkd1a and Pkd1l1[28–30]. We generated double and triple mutants for *pkd1a*, *pkd2*, and *pkd1l1*, and observed valve elongation defects, but no valve cell number differences, in the triple mutants that are similar to those identified in flow-defective zebrafish, with an almost complete absence of the AV valve[6,7,23]. We hypothesize that Pkd1a, Pkd2, and Pkd1l1 are parts of Pkd heteromeric complexes that control cardiac valve formation through blood flow mechanosensation. In this context, it would be interesting to investigate the function of other Pkd2-like proteins, such as Pkd2l1[26], that could interact with Pkd1a or Pkd1l1 during cardiac valve development.

Mutations in *PKD1* are associated with mitral valve regurgitation in humans[24,46]. In addition, *PKD1l1* mutations are linked to atrioventricular septal malformations that are hypothesized to be a consequence of laterality defects[44]. Our experiments suggest that patients

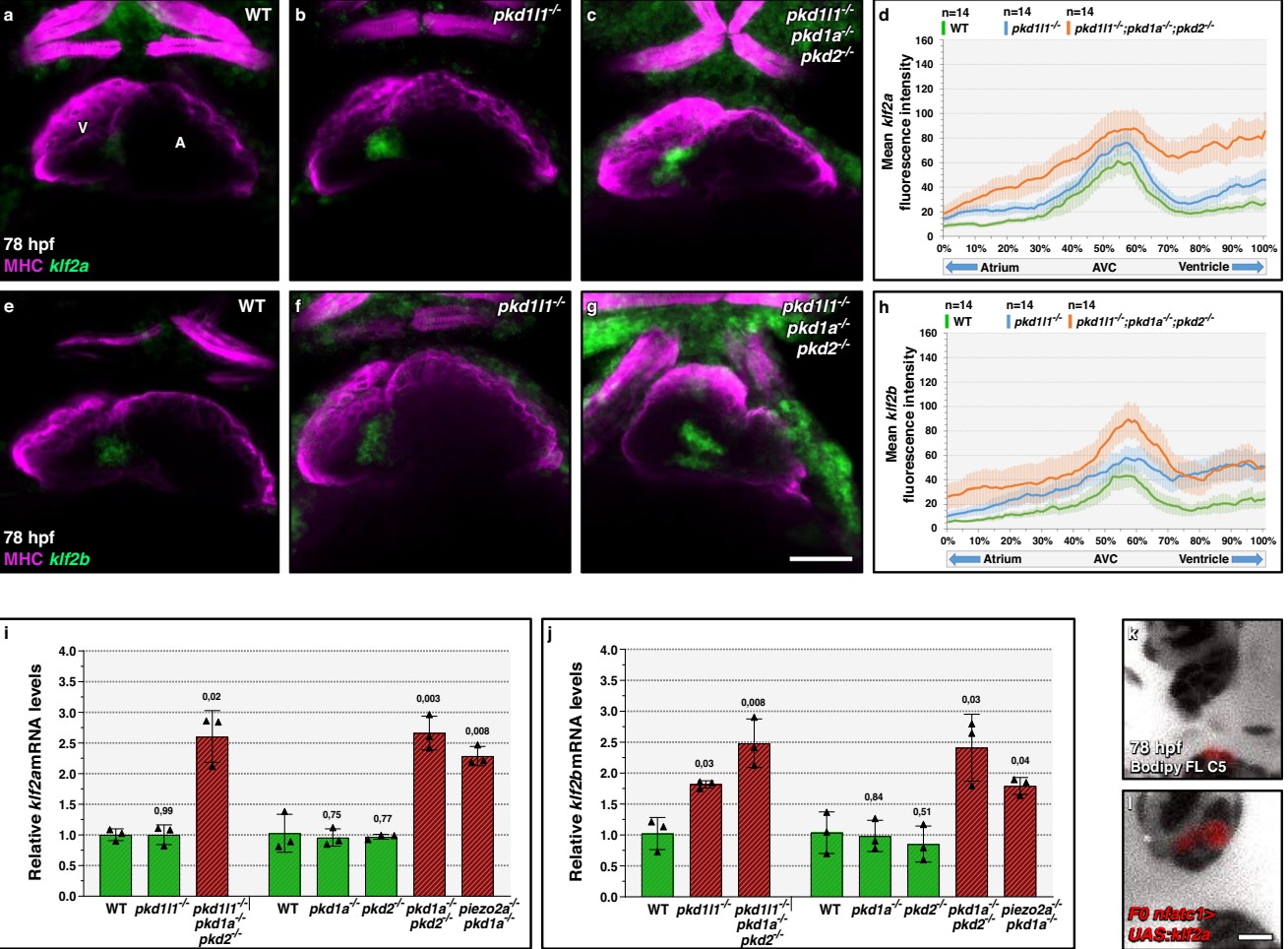

**Fig. 5 | Pkd and Piezo members repress *klf2* expression in the atrioventricular canal. a**–**c**, **e**–**g** Confocal projections of 78 hpf wild-type (**a**, **e**), *pkd1l1* mutant (**b**, **f**), and *pkd* triple mutant (**c**, **h**) hearts stained for MHC, and *klf2a* (**a**–**c**), or *klf2b* (**e**–**g**) expression; ventral views, anterior to the top, ventricle (V) on the left, and atrium (A) on the right. **d**, **h** *klf2a* (**d**) and *klf2b* (**h**) mean fluorescence intensity along the heart tube in 78 hpf wild-type, *pkd1l1* mutant, and *pkd* triple mutant larvae. **i**, **j** Relative mRNA levels of *klf2a* (**i**) and *klf2b* (**j**) in 78 hpf wild-type, *pkd1l1* mutant, *pkd1a* mutant, *pkd2* mutant, *pkd1a; pkd2* double mutant, *pkd* triple mutant, and *piezo2a; pkd1a* double mutant dissected hearts; $n = 3$ biologically independent samples; Ct values are listed in Supplementary Data 1. **k**, **l** Confocal imaging of 78 hpf *nfatc1:Gal4FF* larvae injected at the one-cell stage with a plasmid encoding a *UAS:klf2a-p2a-dTomato*, presenting negative (**k**) or positive (**l**) superior valve cells, and stained with Bodipy FL C5-Ceramide; 32/34 of the negative larvae and 5/13 of the positive larvae display an elongated superior valve; AV canal magnified images. Hearts imaged in ventral view, anterior to the top, ventricle (V) on the left, and atrium (A) on the right. The center of the error bar represents the mean in **i**, **j**. Error bars indicate SEM in **d**, **h**; s.d. in **i**, **j**. *P* values were calculated using a two-sided Student's *t*-test in **i**, **j** and are relative to the wild type. Scale bars: 50 μm in **a**–**c**, **e**–**g**; 10 μm in **k**, **l**. Source data are provided as a Source Data file.

carrying deleterious mutations in *PKD1* or *PKD1L1* may have valvular and atrioventricular defects due to early developmental valve malformations, independently of laterality-related disorders. In addition to their role in AV valve formation, further work will be needed to assess whether *pkd* genes also cooperate during blood flow mechanosensation in other blood vessels endothelial cells[47] or lymphatic vessel endothelial cells[25,48].

Recent studies have identified a role for mechanosensitive channels of the Piezo family in OFT valve development[11,18] and lymphatics valve formation[49]. Here, we investigated the role of two Piezo family members in AV valve formation and found that mutants for *piezo1* and *piezo2a* display valve elongation defects. Furthermore, we also found that Piezo2a cooperates with Pkd1a during cardiac looping and AV valve development. This cooperation between the Piezo and Pkd families appears to be specific to the AV valve, as Pkd2 does not cooperate with Piezo1 during OFT valve formation[11]. Future experiments should address the role of the Pkd family members in OFT valve development and evaluate whether Pkd1a or Pkd1l1 cooperate with Piezo family members in the OFT. The expression of *piezo1* and *piezo2a*

are restricted to the AV canal in zebrafish embryos[18] and larvae (our study). Furthermore, other mechanosensor channels, such as Trpv4, show enrichment in the AV region[50]. Therefore, it would be interesting to generate fluorescent fusion lines of the mechanosensors essential for cardiac valve formation in order to track their trafficking within endocardial cells and study the signals required for their activation[51].

Pkd and Piezo channels have been shown to regulate calcium levels in different tissues[52,53]. In particular, Pkd2 has been described to partially control calcium enrichment in the AV canal, with *pkd2* mutants displaying a similar calcium flux phenotype as the blood flow-defective *tnnt2a* mutants, *cmcl1* morphants, and *gata2* morphants[10]. We found that the *pkd* genes cooperate to control the intracellular calcium levels in the AV canal during different cardiac states. In parallel with the Pkd and Piezo channels, recent work has identified the ATP–dependent purinergic receptor pathway as a blood flow mechanosensor essential for cardiac valve development[12]. This pathway is responsible for triggering calcium pulses in the AV canal upon blood flow stimulation, independently of Pkd2 and Piezo proteins[12]. Therefore, we hypothesize that the generation of AV calcium ion

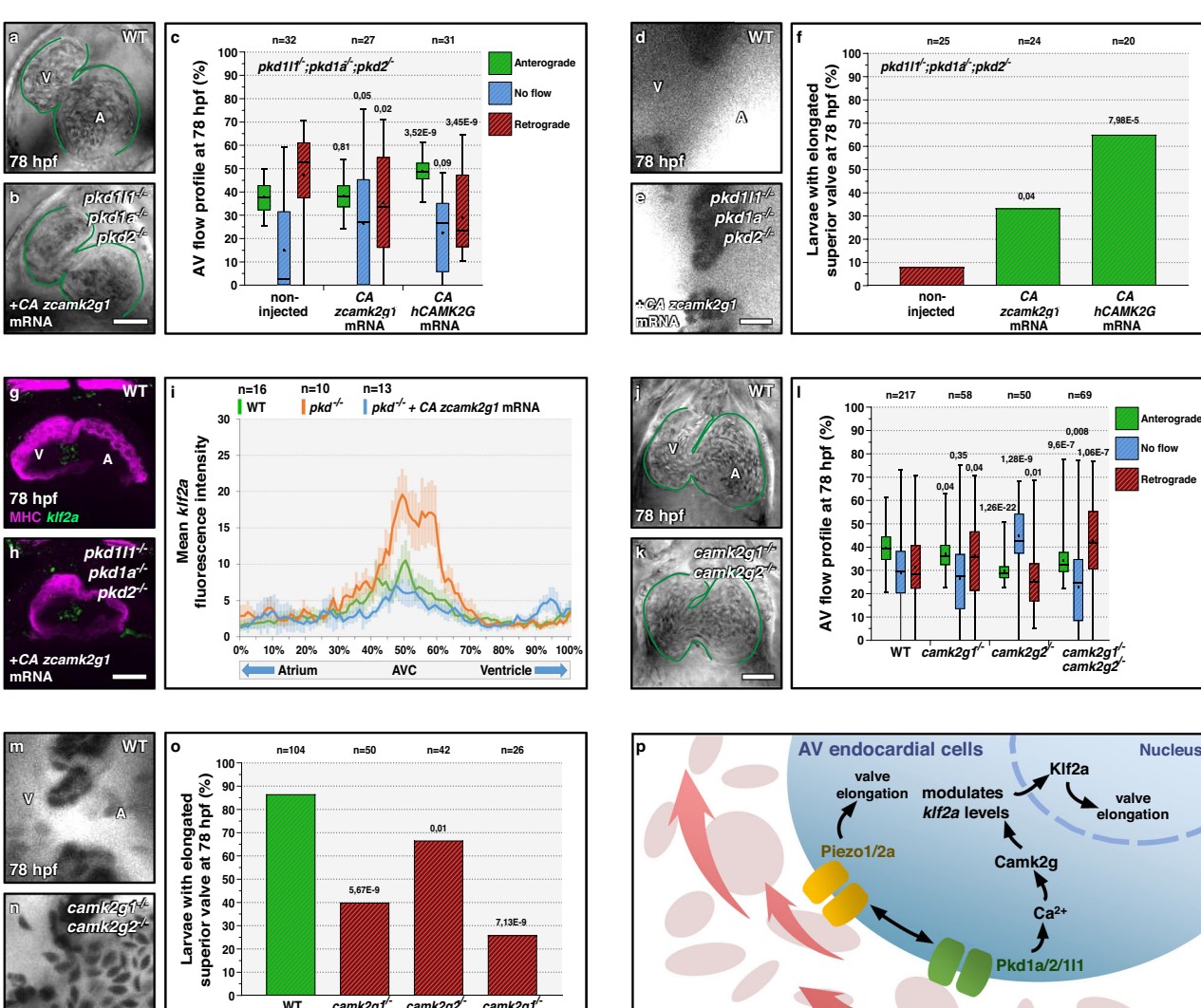

**Fig. 6 | Pkd function in atrioventricular valve development is mediated by Camk2g. a, b** Brightfield images of 78 hpf hearts from non-injected *pkd* triple mutant (**a**) or injected at the one-cell stage with mRNA encoding a CA zCamk2g1 (**b**). **c** AV flow profile of 78 hpf *pkd* triple mutant larvae non-injected or injected at the one-cell stage with mRNA encoding CA zCamk2g1 or CA hCAMK2G. **d, e** Confocal imaging of hearts from 78 hpf non-injected *pkd* triple mutant (**d**) or injected at the one-cell stage with mRNA encoding CA zCamk2g1 (**e**) and stained with Bodipy. **f** Superior AV valve elongation of 78 hpf non-injected *pkd* triple mutants or injected at the one-cell stage with mRNA encoding CA zCamk2g1 or hCAMK2G. **g, h** Confocal projections of hearts from 78 hpf wild-type (**g**) and *pkd* triple mutant injected at the one-cell stage with mRNA encoding CA zCamk2g1 (**h**), stained for MHC and *klf2a* expression. **i** *klf2a* fluorescence intensity along the heart tube in 78 hpf wild types and *pkd* triple mutants non-injected or injected at the one-

cell stage with mRNA encoding CA zCamk2g1. **j, k** Brightfield images of 78 hpf wild-type (**j**) and *camk2g1/2* double mutant (**k**) hearts. **l** AV flow profile of 78 hpf wild-type, *camk2g1* mutant, *camk2g2* mutant, and *camk2g1/2* double mutant larvae. **m, n** Confocal imaging of 78 hpf wild-type (**m**) and *camk2g1/2* double mutant (**n**) hearts stained with Bodipy. **o** Superior AV valve elongation of 78 hpf wild types and *camk2g1/2* mutants. **p** Model summarizing the mechanosensory control of AV valve elongation. Hearts are oriented, annotated, and imaged as in Fig. 1. The box-and-whisker plots in **c, l** represent the same type of data as in Fig. 1g. The center of the bar represents the mean in **f, o**. *P* values are calculated using a two-sided Student's *t*-test in **c, l**; a Fisher's exact test in **f, o**; and are relative to wild type. Error bars indicate SEM in **i**. Scale bars: 50 μm in **a, b, g, h, j, k**; 10 μm in **d, e, m, n**. Source data are provided as a Source Data file.

pulses and the AV calcium enrichment are independent processes, the latter requiring Pkd signaling to occur. In this context, it would be interesting to assess whether double and triple mutants of the channels described in our study would display calcium pulse phenotypes in addition to the enrichment defect.

We also found that the *pkd* genes cooperate to repress the AV endocardial expression of the transcription factor genes *klf2a* and *klf2b*, which are essential for valve formation[7,19,22]. This observation suggests that the expression level of these transcription factors must be tightly regulated and that higher levels of their expression negatively affect AV valve formation. Pkd2 has been reported to positively regulate *klf2a* expression in the OFT[54] and the AV canal[10] based on the quantification of a transgenic reporter. This discrepancy with our

observations could be explained by a differential role of Pkd2 on *klf2a* expression in comparison with other Pkd members or by the different experimental approaches used, which could be investigated in future studies.

Although Klf2a targets have been studied in the myocardium and endocardium during zebrafish valve development[7,19–22], the intermediates of Pkd and Klf2a remained unknown. Here, we identified the *camk2g* genes as upstream mediators of *klf2a* expression and downstream effectors of Pkd proteins during AV valve elongation. CaMKII proteins are serine/threonine kinases[55] involved in a variety of processes during development[38,39,41,56]. CaMKII kinases phosphorylate their target upon stimulation by intracellular calcium elevation[39]. We hypothesize that Pkd-mediated AV calcium enrichment triggers the

activation of Camk2g1 and Camk2g2, which in turn control *klf2a* RNA levels in the endocardial cells. Camk2 proteins have not been described to phosphorylate Klf2a. However, the Ca2+/calmodulin-dependent pathway has been shown to regulate *KLF2* expression in HUVECs, through the phosphorylation of HDAC5 and the transcription factor MEF2[37]. In zebrafish, protein kinase D2 has been shown to control AV valve formation by regulating Hdac5 activity[57]. Moreover, Mef2c overexpression results in aberrant *klf2a* levels in the AV canal[58]. Hence, it will be interesting to determine whether a Camk2g-Hdac5-Mef2c-Klf2a axis is active in controlling AV valve development in zebrafish. It has been previously hypothesized that Camk2g2 compensates for Camk2g1 to control organ laterality[42]. However, we did not observe cardiac laterality defects in *camk2g* double mutants, suggesting that other Camk2 proteins may be involved in this process, and they could also be studied for the control of AV valve formation.

In summary, we propose that members of the *pkd* and *piezo* gene families are essential for AV valve formation, through calcium signaling and the calcium-responsive protein Camk2g.

## Methods

### Zebrafish husbandry

Zebrafish husbandry was performed in accordance with institutional (MPG) and national (German) ethical and animal welfare regulations. Larvae were raised under standard conditions. Adult zebrafish were maintained in 3.5 L tanks at a stock density of 10 zebrafish/L with the following parameters: water temperature: 27–27.5 °C; light/dark cycle: 14/10; pH: 7.0–7.5; conductivity: 750–800 µS/cm. Zebrafish were fed three to five times a day, depending on age, with granular and live food (*Artemia salina*). Health monitoring was performed at least once a year. All embryos and larvae used in this study were raised at 28 °C and staged at 75% epiboly for synchronization. All procedures performed on animals conform to the guidelines from Directive 2010/63/EU of the European Parliament on the protection of animals used for scientific purposes and were approved by the Animal Protection Committee (Tierschutzkommission) of the Regierungspräsidium Darmstadt (reference: B2/1218).

### Crispr/Cas9 mutagenesis

Cas9 mRNA in vitro synthesis was performed according to published protocols[59]. sgRNA design and transcription from an oligonucleotide template were performed as previously described[60]. RNA injection was performed in a wild-type AB background and mutagenesis efficiency was monitored using a T7 endonuclease assay[59]. Mutations were then identified through direct sequencing of PCR products in adult F1 animals. The selected Crispr site sequences for our target genes and primers for T7 endonuclease assays are available in Supplementary Data 1.

### Zebrafish strains and molecular genotyping

The following published lines were used for this study: *klf2a*[bns11 61], *klf2b*[bns12 61], *pkd1a*[hu5855 25], *pkd2*[tc321 31], *tnnt2a*[gbt-R14 62], *Tg(fli1a:Gal4FF)*[ubs4 63], *Tg(UAS:GCaMP6s)*[nkUAShspzGCaMP6s13a 36], and *TgBAC(nfatc1:Gal4)*[mu286 64]. The following mutant lines were newly generated: *pkd1a*[bns507], *pkd1l1*[bns394], *piezo1*[bns340], *piezo2a*[bns367], *camk2g1*[bns508], and *camk2g1*[bns509].

Given the similarity of the phenotypes observed in the two *pkd1a* mutant alleles (*hu5855* and *bns507*), all experiments that mention *pkd1a*[−/−] were performed interchangeably with one of the two alleles. *pkd1a*[bns507] mutants were generated in the *pkd2*[tc321] mutant background because of the proximity of these genes on chromosome 1 (~4 Mb of distance), to facilitate double and triple mutant analysis.

For the experiments with *tnnt2a*, *pkd1a*, or *pkd2* mutants, zygotic mutants from intercrosses of heterozygous parents were used. *tnnt2a* mutants were identified based on the silent heart phenotype at 24 hpf[65], *pkd2* mutants were identified based on their curly tail phenotype at 50 hpf[56], and *pkd1a* mutants were identified after unmounting the

larvae used for bodipy and AV flow profile experiments based on their pericardial edema phenotype at 102 hpf[25]. +/+ and +/− siblings were identified using wild-type and mutant-specific PCR. For the experiments with *piezo1*, *piezo2a*, *pkd1l1*, *camk2g1*, and *camk2g2* mutants, maternal zygotic mutants from an incross of homozygous parents (F3) were used.

For the experiments with *pkd1a; pkd2* double mutants, we intercrossed *pkd1a*[+/−]; *pkd2*[+/−] zebrafish and genotyped the larvae with the *pkd2* curly tail phenotype[66] for the *pkd1a* mutant and wild-type alleles. For the experiments with *pkd1l1; pkd1a* double mutants, we intercrossed *pkd1l1*[−/−]; *pkd1a*[+/−] zebrafish and genotyped the larvae for the *pkd1a* mutant and wild-type alleles. For the experiments with *pkd1l1; pkd2* double mutants, we intercrossed *pkd1l1*[−/−]; *pkd2*[+/−] zebrafish and identified the larvae with the *pkd2* curly tail phenotype[66]. For the experiments with *pkd* triple mutants, we intercrossed *pkd1l1*[−/−]; *pkd1a*[+/−]; *pkd2*[+/−] zebrafish and genotyped the larvae with the *pkd2* curly tail phenotype[66] for the *pkd1a* mutant and wild-type alleles. For the experiments with *piezo2a; pkd1a* double mutants, we intercrossed *piezo2a*[−/−]; *pkd1a*[+/−] or *piezo2a*[+/−]; *pkd1a*[+/−] zebrafish and genotyped the larvae for the *piezo2a* and *pkd1a* mutant and wild-type alleles. For the experiments with *camk2g1; camk2g* double mutants, we intercrossed *camk2g1*[−/−]; *camk2g2*[+/−] zebrafish and genotyped the larvae for the *camk2g2* mutant and wild-type alleles. For the experiments with *klf2a; klf2b* double mutants, we intercrossed *klf2a*[+/−]; *klf2b*[−/−] zebrafish and identified them after unmounting the larvae used for AV flow profile experiments based on their pericardial edema phenotype at 96 hpf[61].

The primers listed in Supplementary Data 1 were used to genotype by PCR the mutant and wild-type alleles described in this study. PCR amplifications were performed with KAPA2G Fast Ready Mix (Sigma) using the following cycling conditions: Initial denaturation 3 min 95 °C; 10 cycles (15 s 95 °C, 30 s 65 °C first to 55 °C last cycle, 30 s 72 °C); 20 cycles (15 s 95 °C, 30 s 55 °C, 30 s 72 °C); final extension 1 min 72 °C.

### Plasmid generation

The *camk2g1* and *klf2a* ORF were amplified from a pool of 78 hpf larval cDNA, and cloned into a pCS2+ vector. The T287D mutation was introduced in *zcamk2g1*-pCS2+ through PCR amplification. *hCAMK2G*[T287D] was amplified from a β-actin:GFP- *hCAMK2G*[T287D] vector[41], and cloned into a pCS2+ vector. *klf2a* was amplified from *klf2a*-pCS2+ and cloned into a 5XUAS:p2a-dTomato vector. All primers are listed in Supplementary Data 1, and vector ligation was performed using the in vivo cloning method[67].

### RNA and morpholino injections

*zcamk2g1*[T287D]-pCS2+ and *hCAMK2G*[T287D]-pCS2+ were linearized using NotI, and mRNAs were in vitro synthesized using the mMessage mMachine SP6 transcription kit (Ambion). We injected 2 ng of the *tnnt2a* morpholino[65], 5 ng of the *pkd1a* morpholino (ex8)[45], 50 pg of *zcamk2g1*[T287D] or *hCAMK2G*[T287D] mRNAs, and 13 pg of *5XUAS:klf2a-p2a-dTomato* plasmid DNA together with 13 pg of transposase mRNA. All reagents were injected at the one-cell stage at a volume of 1 nl together with 0.2% phenol red.

### RT-qPCR analysis

For every condition, RNA was extracted from a pool of 15 dissected hearts (Fig. 5i, j) or from a pool of 10 larvae (Figs. S2b, S3b). RNA was isolated using TRIzol extraction and reverse transcription was performed using the Maxima First Strand cDNA synthesis kit (Thermo Fisher). The results represent biological triplicates of dissected hearts or larvae and two technical duplicates were performed per biological replicate. *eef1b2* was used as a reference gene for all conditions. Fold changes were calculated using the $2^{-\Delta\Delta Ct}$ method. The mutant conditions were compared with AB wild types. Animals were genotyped prior to heart dissection (Fig. 5i, j) or RNA extraction from whole larvae

(Figs. S2b, S3b). All Ct values and primers are listed in Supplementary Data 1.

## RNA in situ hybridizations procedure

Whole-mount RNA in situ hybridizations were performed according to standard protocols[68]. The in situ probes for *piezo1*, *piezo2a*, *klf2a*, and *klf2b* were synthesized directly from a PCR product, performed on wild-type AB cDNA at 78 hpf, using an incorporated T7 promoter (Supplementary Data 1). For the FISH experiments, the same protocol was used, with the following modifications: During the first day, larvae were incubated in 1% H2O2 for 30 min. During the second day, larvae were incubated with the primary antibodies anti-DIG-POD (1:5000, Roche 11207733910) and anti-MHC (MF20) (1:500, Invitrogen 14-6503-82) diluted in PBST + 10% Western Blocking reagent (Roche 11921673001). On the third day, after six washes in PBST, larvae were incubated with an amplification solution for 2 min (Akoya Biosciences NEL741001KT). Fluorescein was then added at 1:100 to the amplification solution, then kept in the dark at room temperature for 20 min. The larvae were then incubated for 2 h with the secondary antibody Goat anti-Mouse-Alexa647 (1:500, Invitrogen A21236), and extensively washed in PBST before imaging. In situ hybridizations were imaged using an SMZ25 stereomicroscope (Nikon) with a 2x/0.3 objective (Fig. S3k, l), an LSM 700 confocal laser scanning microscope (Zeiss) with a 40x objective (Fig. 6g, h, Fig. S5a–s), and an LSM 800 Examiner confocal laser scanning microscope (Zeiss) with a 40x objective (Fig. 5a–h). For the FISH experiments, the images represent a maximal projection of ten z-planes (20 μm). The experiments presented in Fig. 5a–h and S5d–s were performed using the same crosses as for the RT-qPCR analysis shown in Fig. 5i, j.

## Immunohistochemistry

Whole-mount immunostaining was performed as described previously[69], with the following modification: Larvae were fixed in 4% PFA after stopping the heart with 0.2% Tricaine to prevent cardiac collapse during fixation. The primary antibodies used in this study are Mouse anti-MHC (MF20) (1:500, Invitrogen 14-6503-82), Mouse anti-Alcama (1:500, ZN-8, DSHB), and Rabbit anti-Fli1 (1:500, Abcam ab133485), followed by the secondary antibodies: Goat anti-Mouse-Alexa568 (1:500, Invitrogen A11004), Goat anti-Mouse-Alexa647 (1:500, Invitrogen A21236), and Goat anti-Rabbit-Alexa488 (1:500, Invitrogen A110034). All images were analyzed using an LSM 700 confocal laser scanning microscope (Zeiss) with a 40x objective, except for the samples shown in Fig. 2t, u, which were imaged using an LSM 800 Examiner confocal laser scanning microscope (Zeiss) with a 40x objective. For AV cell number quantification, the cells were counted in every plane of a z-stack covering the AV canal. The 78 hpf wild-type dataset (*n* = 61) is the same in Figs. 1n, 2s, 3o, and represents a pool of *pkd1a*⁺/⁺, *pkd1a*⁺/⁺; *pkd2*⁺/⁺, and *camk2g1*⁺/⁺; *camk2g2*⁺/⁺ larvae. The 78 hpf *pkd1a* mutant dataset (*n* = 26) is the same in Figs. 1n, 2s, 3o.

## AV flow profile and heart rate quantification

Zebrafish hearts were imaged live using an inverted Cell Observer Spinning Disk microscope with a 25x objective at 240 fps with a pixel width of 1.92 μm. The AV blood flow profile was analyzed using a custom-made ImageJ script that calculates the percentage of anterograde (from the atrium to the ventricle), retrograde (from the ventricle to the atrium), or no-flow (no movement between the atrium and the ventricle) blood movement in a beating cycle. The results were then averaged between 3 beating cycles per embryo/larva. The profiles are represented using a box-and-whisker plot, which depicts the data through their quartiles. The 78 hpf wild-type dataset (*n* = 217) is the same in Figs. 1g, h, 2e, 3e, 6l, and represents a pool of *pkd1a*⁺/⁺, *pkd1a*⁺/⁺; *pkd2*⁺/⁺, *klf2a*⁺/⁺; *klf2b*⁺/⁺, and *camk2g1*⁺/⁺; *camk2g2*⁺/⁺ larvae. The same

dataset was used to measure the heart rate (Figs. S1i, S2j, S3j, S6d). The 78 hpf *pkd1a* mutant dataset (*n* = 51) is the same in Figs. 1h, 2e, 3e.

## In vivo Bodipy labeling and imaging

Bodipy FL C5-Ceramide (Invitrogen D3521) was added to fish media 1 day prior to imaging at 0.2 μM from a 1 mM stock solution. The larvae were anesthetized in 0.0175% tricaine and their heart stopped using 25 mM BDM (Sigma Aldrich B0753) treatment. Images were then acquired using an LSM 700 confocal laser scanning microscope (Zeiss) using a 40x objective, and the presence of a superior valve leaflet was assessed in every plane of a z-stack covering the AV canal. The 78 hpf wild-type dataset (*n* = 104) is the same in Figs. 1k, 2l, 3j, 6o, and represents a pool of *pkd1a*⁺/⁺, *pkd1a*⁺/⁺; *pkd2*⁺/⁺, and *camk2g1*⁺/⁺; *camk2g2*⁺/⁺ larvae. The 78 hpf *pkd1a* mutant dataset (*n* = 32) is the same in Figs. 1k, 2l, 3j.

## 4D live imaging of GCamPs calcium sensor

*fli1a:Gal4FF; UAS:GCamP6s* positive larvae were selected under a fluorescence stereomicroscope. Time-lapse imaging was acquired using a spinning disc microscope at 50 fps with a pixel width of 1.92 μm. Fifty acquisitions were performed for every larva, and the optical plane was moved by 4 μm between movies to cover a z-section depth of 200 μm per heart, starting ventrally. The 50 acquisitions were temporally realigned using a custom-made ImageJ script. We then performed a maximal projection of the z-dimension using ImageJ to visualize the GCamP6s signal in the whole heart, at the atrial diastole and systole states.

## Heart segmentation

Heart segmentation was performed on three types of movies: the imaging files generated to measure the AV flow profile (Fig. S1b–e′), the GCamPs 4D movies (Fig. S4a–c′), and the fluorescence in situ z-stack images (Fig. S5a, b). Using a custom-made ImageJ script, we trace two external lines, one on each side of the heart, from the inflow to the outflow, and indicate the AV canal. The script then traces a midline of the heart, calculated to be at mid-distance from the two external lines. The midline tracing is performed independently in the atrium and the ventricle in order to account for the differences in their curvature. The heart is then segmented along the midline by 100 segments, which we use to extract heart diameter and fluorescence intensity parameters from our images. The 78 hpf wild-type (*n* = 20) and *tnnt2a* mutant (*n* = 18) datasets are the same in Figs. S1g, h, S2h, i, S3h, i, S6b, c, and the wild types represent a pool of *pkd1a*⁺/⁺, *pkd1a*⁺/⁺; *pkd2*⁺/⁺, and *camk2g1*⁺/⁺; *camk2g2*⁺/⁺ larvae.

## Software, statistics, and reproducibility

Microscopy images were acquired with NIS-Elements 4.30, ZEN 2011, ZEN 2.5 Blue, and ZEN. 3.5 Blue. RT-qPCR data were acquired with Bio-Rad CFX Manager 3.1. AV flow profile analysis, valve elongation quantification, and heart segmentation were performed using Fiji (ImageJ 1.53q). Sequence analysis was performed using ApE (v2.0.54). Data were processed with GraphPad Prism 9 and Microsoft Office 2016. A two-sided Fisher exact test was used to calculate *p* values for the bodipy experiments and a two-sided Student's *t*-test was used to calculate *p* values for the other experiments. Experiments were performed at least three times independently.

## Reporting summary

Further information on research design is available in the Nature Portfolio Reporting Summary linked to this article.

## Data availability

Materials and raw images that support the findings of this study are available upon request to the corresponding authors. Raw data are

provided in a Source Data file. Source data are provided with this paper.

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

## Acknowledgements

We thank F. Gunawan for discussions and comments on the manuscript, S. Howard and H-M Maischein for technical support, J. Vermot for providing the *pkd2^{tc321}* mutant line, B.M. Hogan for providing the *pkd1a^{hu5855}* mutant line, K. Kawakami for providing the *Tg(UAS:G-CamP6s)^{nkUAShspzGCaMP6s13a}* line, and S.C. Rothschild and R.M. Tombes for providing the *β-actin:GFP-hCAMK2G^{T287D}* vector. This work was supported by a European Molecular Biology Organization Long-Term Fellowship (ALTF 1234-2018) to T.J. and funds from the Max Planck Society to D.Y.R.S.

## Author contributions

Conceptualization (T.J. and D.Y.R.S.), Methodology (T.J.), Investigation (T.J., A.R.d.S., B.C., S.L., and V.C.), Resources (D.Y.R.S.), Writing (T.J. and D.Y.R.S. with inputs from A.R.d.S., B.C., S.L., and V.C.), Supervision (D.Y.R.S.), Project administration and funding acquisition (T.J. and D.Y.R.S.).

## Funding

## Competing interests

The authors declare no competing interests.
