## [Peer Review File · Nature Communications]

Multiple pkd and piezo gene family members are required for atrioventricular valve formationREVIEWER COMMENTS

Reviewer #1 (Remarks to the Author):

Juan and colleagues have delineated the role of Pkd1a, Pkd2 and Pkd1l1 in an elongation process by which atrioventricular valve leaflets of zebrafish elongate. They report that these mechanosensory proteins repress the expression of Klf2a, Klf2b and Egr1 transcription factors. The authors also show that Piezo1 and Piezo2b are expressed in the AV region and act along with Pkd in AV valve leaflet elongation. Furthermore, they show that Calcium-dependent kinases Camk2g1 and Camk2g2 act downstream of Pkd. The study presents interesting insights about the involvement of Pkd family mechanosensors in AV valvulogenesis and valve leaflet elongation. The manuscript is clearly written and figures are mostly of good quality. However, there are still a number of issues to some of the data presented in this study that are summarized below.

Major comments:

- 1) The expression patterns of pkd1a, pkd2 and pkd1l1 within the heart have not been shown. Whole mount in situ hybridizations experiments are needed to confirm whether these genes are expressed at the atrioventricular canal at the time points mentioned in the study. Alternatively, knockdown of these genes may impact the vasculature and blood flow in a way that would have a more indirect effect on valvulogenesis. In this context it would be good to also depict the AV flow profile for pkd2 and pkd1l1 single mutants.
- 2) The cardiac phenotype shown in Figure 1 d-f has been reported previously (Coxam et al., 2014) and needs to be acknowledged.
- 3) Is there a reduction in cell proliferation or, alternatively, does apoptosis occur in AV cells of pkd1a mutants? Is that causing a reduction in cell numbers?
- 4) The manuscript did not report on the numbers of AV cells in pkd double and triple mutants and in piezo mutants? This would be important to assess their roles in this process as well. Also, it will be important to assess this phenotype as early as possible: Are AV cell numbers affected before 78hpf (i.e. at 54hpf) in pkd mutants?
- 5) Is the pkd phenotype a complete failure of valve leaflet elongation or merely a delay? Can the authors provide images of the AV valves at later time points (i.e. 96hpf)?
- 6) It is a bit puzzling that the pkd2 and pkd1l1 mutants are reported to show looping defects, as shown previously, but that 'double and triple mutant combinations of the pkd genes lead to a wild-type-like heart morphology.' Can the authors explain this discrepancy?
- 7) If the pkd mechanosensors are cooperating in affecting valve elongation, would restoring the expression of one or two of the pkd in a triple mutant bring about any rescue to the phenotype?
- 8) How significant is the increase in retrograde flow in triple mutants in comparison to pkd1a mutants (Fig 2g)?
- 9) Fig 4-g, the fluorescence intensity graph and the image depicted above don't seem to correlate. It appears that triple mutants have higher intensity in the ventricle compared with WT. Also tnnt2a morphants seems to have much lower calcium levels (as expected) than WT although the graph suggests a higher basal value of calcium levels.
- 10) The most significant decrease in valve elongation occurs in pkd1a and piezo2 double mutants. How does this correlate with the expression of klf2a and klf2b? Similarly, how is the expression of klf2a and klf2b in pkd1a and pkd2 single mutants and in double mutants? Can the authors provide this information which will be important to assess the relevance of these downstream factors for the phenotypes (qPCR and in situ).
- 11) Fig 1h shows pkd1a mutants have an increased retrograde flow, which is similar to that in klf2 mutants. However, fig5 shows that klf2a and klf2b levels are elevated in pkd triple mutants. How do the authors account for this discrepancy?
- 12) How are the luminal versus abluminal populations of AV cells distributed in pkd mutants? A counterstaining against Alcam may provide more insight. This will also be important for appreciating the way valve leaflets are elongating.
- 13) Injection of Camk2g1CA mRNA into pkd mutants does not significantly rescue the flow profile (Fig.

6a-c). Similarly, *camk2g1* mutants have an AV flow profile similar to WT. Also, injection of *camk2g1*WT mRNA does not rescue the valve elongation defect in *pkd* mutants. Hence, claiming that *camk2g1* acts downstream of *pkd* in mediating its effect in AV valve formation is not well substantiated.

Minor comments:

- 1) The statistical significance for the fractions of *pkd1a* mutant or *pkd* double mutants showing defects in valve elongation is missing (Fig. 1k).
- 2) The labelling of . fig. 2k-m is hiding parts of the images.

Reviewer #2 (Remarks to the Author):

The authors identified a novel pathway and molecular mechanisms for the zebrafish AV valve formation. While the results are interesting and the techniques are compelling, the following comments would clarify and strengthen the conclusion.

1. Is there a genetic compensation for *pkd1a* mutant alleles since the AV valve phenotype is mild? How is expression of other genes (*pkd2*, *pkd1l1*, *piezo1*, and *piezo2*) affected in *pkd1a* mutants?
2. It is not clear how Piezo and Pkd interact with each other to regulate calcium levels. Can piezo mutants or *piezo2a*; *pkd1a* double mutant phenotypes be rescued by expressing downstream genes *camk2g* or by *klf2a/b* KD? Furthermore, can activating Piezo suppress *camk2g1* phenotypes?
3. The authors stated that every mutant combination that involves *pkd2* or *pkd1|1* leads to randomized cardiac looping. While the authors demonstrated that *pkd1* MO mediated the looping defects in *piezo2a* mutants, unclear is whether *pkd2* or *pkd1|1* modulates the similar effects.

Reviewer #1 Comments:

Juan and colleagues have delineated the role of Pkd1a, Pkd2 and Pkd11 in an elongation process by which atrioventricular valve leaflets of zebrafish elongate. They report that these mechanosensory proteins repress the expression of Klf2a, Klf2b and Egr1 transcription factors. The authors also show that Piezo1 and Piezo2b are expressed in the AV region and act along with Pkd in AV valve leaflet elongation. Furthermore, they show that Calcium-dependent kinases Camk2g1 and Camk2g2 act downstream of Pkd. The study presents interesting insights about the involvement of Pkd family mechanosensors in AV valvulogenesis and valve leaflet elongation. The manuscript is clearly written and figures are mostly of good quality. However, there are still a number of issues to some of the data presented in this study that are summarized below.

We thank the reviewer for their supportive comments

Major comments:

1) The expression patterns of pkd1a, pkd2 and pkd11 within the heart have not been shown. Whole mount in situ hybridizations experiments are needed to confirm whether these genes are expressed at the atrioventricular canal at the time points mentioned in the study. Alternatively, knockdown of these genes may impact the vasculature and blood flow in a way that would have a more indirect effect on valvulogenesis. In this context it would be good to also depict the AV flow profile for pkd2 and pkd11 single mutants.

We fully agree that the knockdown of the genes presented in this study may have an indirect effect on valvulogenesis. However, only a tissue specific knockdown using the Cre-Lox system or targeted Crispr/Cas9 would answer this question precisely, which are experiments that go beyond the scope of this work. Recent work has shown that *pkd2* is expressed homogeneously in the endocardium as early as 32 hpf using RNA scope (Vignes et al, 2022), and that *pkd1a* is expressed in the endocardium at 5 dpf (bulk RNA-seq on sorted cells, unpublished data from our laboratory). These mechanosensors are hypothesized to be present in the whole endocardium and respond locally to flow differences (i.e. high shear stress and oscillatory flow in the atrioventricular canal region). Therefore, we do not expect to find them enriched in the atrioventricular canal. We have added the atrioventricular flow profile for *pkd2* and *pkd11* single mutants (Fig. 2e).

2) The cardiac phenotype shown in Figure 1 d-f has been reported previously (Coxam et al., 2014) and needs to be acknowledged.

We have modified the text accordingly.

3) Is there a reduction in cell proliferation or, alternatively, does apoptosis occur in AV cells of pkd1a mutants? Is that causing a reduction in cell numbers?

After evaluating the reviewer's question about the number of atrioventricular canal cells in *pkd* double and triple mutants (see below), we also increased the 'n' value in *pkd1a* single mutants. This work led us to revise our claim that *pkd1a* mutants display fewer atrioventricular canal cells, and we have now updated Fig. 1n showing that *pkd1a* mutants do not present significant differences in atrioventricular cell number compared with wild type. Therefore, we did not assess the reduction in cell number using proliferation or apoptosis assays.

4) The manuscript did not report on the numbers of AV cells in pkd double and triple mutants and in piezo mutants? This would be important to assess their roles in this process as well. Also, it will be

important to assess this phenotype as early as possible: Are AV cell numbers affected before 78hpf (i.e. at 54hpf) in pkd mutants?

We have now determined the number of atrioventricular canal cells in *pkd* double and triple mutants, as well as in *piezo* mutants. We did not find significant differences in atrioventricular cell number in any mutant combinations. Therefore, we did not assess the onset of this phenotype at 54 hpf.

5) Is the pkd phenotype a complete failure of valve leaflet elongation or merely a delay? Can the authors provide images of the AV valves at later time points (i.e. 96hpf)?

As presented in Fig. 1d-f', *pkd1a* mutant hearts are collapsed at 102 hpf, and the blood does not flow normally through the heart. This phenotype is already present at 96 hpf, and the valve defects observed at this stage are therefore likely the consequences of blood flow defects.

6) It is a bit puzzling that the pkd2 and pkd11 mutants are reported to show looping defects, as shown previously, but that 'double and triple mutant combinations of the pkd genes lead to a wild-type-like heart morphology.' Can the authors explain this discrepancy?

We agree with the reviewer that the word choice "wild-type-like heart morphology" does not accurately represent the data, and we have now replaced it with "wild-type-like cardiac chamber size".

7) If the pkd mechanosensors are cooperating in affecting valve elongation, would restoring the expression of one or two of the pkd in a triple mutant bring about any rescue to the phenotype?

We were unable to clone *pkd11* and *pkd1a* given the size of their coding region (7.7 kb and 13 kb respectively). However, attempts to rescue *pkd* triple mutant phenotype using *pkd2* mRNA injections resulted in early left-right asymmetry defects and severe cardiac defects, which prevented the interpretation of valve phenotypes. We therefore decided not to include this experiment in the manuscript.

8) How significant is the increase in retrograde flow in triple mutants in comparison to pkd1a mutants (Fig 2g)?

We have now included the significant ($p=0.03$) difference between *pkd1a*; *pkd2* double mutants and *pkd* triple mutants, in comparison to the non-significant ($p=0.46$) difference between *pkd1a* mutants and *pkd1a*; *pkd2* double mutants in Fig. 2e.

*9) Fig 4-g, the fluorescence intensity graph and the image depicted above don't seem to correlate. It appears that triple mutants have higher intensity in the ventricle compared with WT. Also *tnnt2a* morphants seems to have much lower calcium levels (as expected) than WT although the graph suggests a higher basal value of calcium levels.*

We agree with the reviewer, and have now selected more representative images of *pkd* triple mutants for this figure.

Regarding the *tnnt2a* morphants, we observed that they display a decrease in calcium levels in the atrioventricular canal specifically while the basal calcium levels remain high.

10) The most significant decrease in valve elongation occurs in pkd1a and piezo2 double mutants. How does this correlate with the expression of klf2a and klf2b? Similarly, how is the expression of klf2a and klf2b in pkd1a and pkd2 single mutants and in double mutants? Can the authors provide

this information which will be important to assess the relevance of these downstream factors for the phenotypes (qPCR and in situ).

We have now performed fluorescence *in situ* hybridization and RT-qPCR for *klf2a* and *klf2b* in *pkd1a* and *pkd2* single and double mutants, as well as in *pkd1a; piezo2a* double mutants; these new data are shown in Fig. 5 and Fig. S5.

11) Fig 1h shows *pkd1a* mutants have an increased retrograde flow, which is similar to that in *klf2* mutants. However, fig5 shows that *klf2a* and *klf2b* levels are elevated in *pkd* triple mutants. How do the authors account for this discrepancy?

We injected a *UAS:klf2a-p2a-dTomato* plasmid in *nfatc1:Gal4FF* positive embryos to overexpress *klf2a* in atrioventricular valve endocardial cells and found that the F0 larvae presenting a positive dTomato signal in the atrioventricular canal region display a reduction in valve elongation compared to the atrioventricular canal negative larvae. We propose from these observations that *klf2a* levels are tightly regulated to promote valve elongation: reduction or increase in *klf2a* expression leads to similar effects.

12) How are the luminal versus abluminal populations of AV cells distributed in *pkd* mutants? A counterstaining against *Alcama* may provide more insight. This will also be important for appreciating the way valve leaflets are elongating.

We performed *Alcama* immunostaining in *pkd* triple mutants and observed that luminal and abluminal populations of valve cells appeared to be unaffected (Fig. 2t-u). Only the elongation appears to be specifically disrupted in *pkd* triple mutants as suggested by our live Bodipy staining (Fig. 2f-l).

13) Injection of *Camk2g1CA* mRNA into *pkd* mutants does not significantly rescue the flow profile (Fig. 6a-c). Similarly, *camk2g1* mutants have an AV flow profile similar to WT. Also, injection of *camk2g1WT* mRNA does not rescue the valve elongation defect in *pkd* mutants. Hence, claiming that *camk2g1* acts downstream of *pkd* in mediating its effect in AV valve formation is not well substantiated.

We have now performed a more convincing analysis by using a CA version of human CAMK2G, which has a high amino acid sequence similarity to both zebrafish *Camk2g1* (81.29 ID%, ensembl) and zebrafish *Camk2g2* (85.20 ID%, ensembl), which is likely to increase its potency in zebrafish. The resulting data are shown in Fig. 6a-f.

Minor comments:

1) The statistical significance for the fractions of *pkd1a* mutant or *pkd* double mutants showing defects in valve elongation is missing (Fig. 1k).

We have now added the p-values in the valve elongation figures, and increased the 'n' value for the *pkd1a* mutant experiment to strengthen the associated p-value.

2) The labelling of . fig. 2k-m is hiding parts of the images.

We have changed the cropping of this figure to show the phenotype more clearly.

Reviewer #2 Comments:

The authors identified a novel pathway and molecular mechanisms for the zebrafish AV valve formation. While the results are interesting and the techniques are compelling, the following comments would clarify and strengthen the conclusion.

We thank the reviewer for their supportive comments

*1) Is there a genetic compensation for *pkd1a* mutant alleles since the AV valve phenotype is mild? How is expression of other genes (*pkd2*, *pkd1l1*, *piezo1*, and *piezo2*) affected in *pkd1a* mutants?*

We have performed RT-qPCR for *pkd2*, *pkd1l1*, *piezo1*, and *piezo2a* in *pkd1a* mutants, and observed that *pkd1l1* is upregulated in *pkd1a* mutants, while the others are unchanged (Fig. S2b, Fig. S3b).

*2) It is not clear how Piezo and Pkd interact with each other to regulate calcium levels. Can piezo mutants or *piezo2a*; *pkd1a* double mutant phenotypes be rescued by expressing downstream genes *camk2g* or by *klf2a/b* KD? Furthermore, can activating Piezo suppress *camk2g1* phenotypes?*

We failed to rescue the *pkd1a*; *piezo2a* mutant phenotype using CA human CAMK2G or CA zebrafish Camk2g1. We propose from these data that the role of Piezo in valve elongation is mediated by a pathway other than Camk2g. Therefore, we did not perform the reviewer's suggested experiment to try and suppress Camk2g phenotypes using the Piezo1 agonist drug Yoda1. In addition, our experiments suggest that *pkd1a* interacts with *piezo2a*, but not *piezo1*, making the interpretation of the results of such an experiment uncertain.

*3) The authors stated that every mutant combination that involves *pkd2* or *pkd1/1* leads to randomized cardiac looping. While the authors demonstrated that *pkd1* MO mediated the looping defects in *piezo2a* mutants, unclear is whether *pkd2* or *pkd1/1* modulates the similar effects.*

Whereas *pkd1a* morphants, or mutants, do not display cardiac looping defects, *pkd2* and *pkd1l1* mutants display completely randomized cardiac looping. Therefore, one does not expect to find an increased incidence in looping defects in *piezo2a*; *pkd2/pkd1l1* double mutants compared with *pkd2* or *pkd1l1* single mutants.

REVIEWERS' COMMENTS

Reviewer #1 (Remarks to the Author):

The manuscript has been much improved. The authors have addressed almost all questions that were raised. I would like to congratulate the authors and suggest that the manuscript can be accepted in its present form.